# Effects of Fat Supplementation in Dairy Goats on Lipid Metabolism and Health Status

**DOI:** 10.3390/ani9110917

**Published:** 2019-11-04

**Authors:** Giovanni Savoini, Fabio Omodei Zorini, Greta Farina, Alessandro Agazzi, Donata Cattaneo, Guido Invernizzi

**Affiliations:** Department of Health, Animal Science and Food Safety “Carlo Cantoni”, University of Milan, 20133 Milan, Italy; giovanni.savoini@unimi.it (G.S.); fabio.omodei@unimi.it (F.O.Z.); greta.farina85@gmail.com (G.F.); alessandro.agazzi@unimi.it (A.A.); donata.cattaneo@unimi.it (D.C.)

**Keywords:** transition dairy goat, polyunsaturated fatty acids, lipid metabolism, immune response

## Abstract

**Simple Summary:**

There is an increasing demand for information on the nutraceutical properties of food. Due to its bioactive components and high digestibility, goat milk is an excellent functional food. Dietary fat supplementation can further enrich the value of goat milk by modifying its acidic profile. Nevertheless, animal health can also benefit from lipids supplied with rations. In this review, the relationships between dietary fats and goat health status are summarized. Particular attention is paid to describing the effects of specific fatty acids on lipid metabolism and immune functionality.

**Abstract:**

Fat supplementation has long been used in dairy ruminant nutrition to increase the fat content of milk and supply energy during particularly challenging production phases. Throughout the years, advances have been made in the knowledge of metabolic pathways and technological treatments of dietary fatty acids (FAs), resulting in safer and more widely available lipid supplements. There is an awareness of the positive nutraceutical effects of the addition of eicosapentaenoic acid (EPA) and docosahexaenoic acid (DHA) to fat supplementation, which provides consumers with healthier animal products through manipulation of their characteristics. If it is true that benefits to human health can be derived from the consumption of animal products rich in bioactive fatty acids (FAs), then it is reasonable to think that the same effect can occur in the animals to which the supplements are administered. Therefore, recent advances in fat supplementation of dairy goats with reference to the effect on health status have been summarized. In vivo trials and in vitro analysis on cultured cells, as well as histological and transcriptomic analyses of hepatic and adipose tissue, have been reviewed in order to assess documented relationships between specific FAs, lipid metabolism, and immunity.

## 1. Introduction

In recent years, the identification of various biologically active components in ruminant milk and dairy products, coupled with the growing attention of general consumers with respect to the nutraceutical and health properties of food, has generated interest on their potential beneficial effects on human health [1]. Goat milk has a high nutritional value and a large number of bioactive components [2]. Goat milk is also more easily digestible than bovine milk, mostly due to the higher β-casein fraction, smaller diameter of fat globules, and the absence of agglutinins [3,4]. All these features make goat milk a particularly appealing ‘functional food’.

Dietary manipulation of milk fatty acid (FA) composition represents an interesting approach to further improve the biological and nutraceutical properties of dairy goats, with beneficial effects on both human and animal health. Generally, feedstuffs in the ruminant diet have a low content in lipids, most of which are unsaturated in nature (unsaturated fatty acids, UFAs). In particular, forages are rich in α-linolenic (C18:3n-3) and linoleic (C18:2n-2) acids [5], precursors of the n-3 and n-6 polyunsaturated fatty acid (PUFA) lines, respectively. Conversely, cereals and oilseeds—mainly used to increase the ration energy and lipid content—are richer in linoleic acid and oleic acid (C18:1n-9), a monounsaturated fatty acid (MUFA) widely represented in dairy milk fat [6]. In contrast to what could be expected, the FA profile of milk fat shows large differences from the dietary one. In fact, at least 60%–70% of dairy milk fat is composed of saturated fatty acids (SFAs), with a remaining 20%–35% MUFAs and only up to a 5% content in PUFAs [7]. These differences can be largely attributed to the rumen biohydrogenation process. During the biohydrogenation process, a large fraction of dietary UFAs—ranging from 60% to 90% [8,9]—undergo hydrolysis and several consecutive isomerization reactions that lead to their saturation. By consequence, the main FAs that reach the duodenum are stearic acid (C18:0) and various intermediates of reaction, mostly trans-isomers of oleic and linoleic acids [10,11]. PUFAs found in milk fat derive from the small amount of rumen-escaped PUFAs and—in a minor part—by the enzymatic activity of mammary Δ9 desaturase [12].

Nevertheless, it is undoubted how nutritional factors (forage type, forage/concentrate ratio, fat sources) and feed processing can still easily affect the FA profile of milk. It has been proved that dairy animals fed fresh forages and grazing grass produce milk higher in PUFAs compared to fodder-fed animals [13,14], and that the type of pasture can affect FA profile [15]. Variations of milk FA content were also reported as a consequence of different forage to concentrate ratios [16,17] as well as management and farming systems [18,19]. Likewise, lipid supplements have proved to be effective in modulating the acidic composition of milk [20,21,22,23]. In recent years several efforts have been made to increase the n-3 PUFA content of milk in both cows and goats. While n-3 PUFAs have the potential to improve long-term human health [24], unfortunately mammals are unable to synthetize them, a reason for which they necessarily have to be integrated through the diet [25]. Long chain derivates of C18:3 n-3, eicosapentaenoic acid (EPA) and docosahexaenoic acid (DHA), are the two main essential n-3 PUFAs. Due to their poor content in traditional ruminant diets, EPA and DHA are very scarcely represented in dairy milk fat (<0.1% of total FAs) [26]. A possibility to enrich dairy ruminant milk with EPA and DHA is to provide feedstuffs high in them (fish oil, marine algae) or high in their precursor C18:3 n-3 (linseed). Common to dietary long-chain PUFAs, regardless of their source, is their low milk transfer rate [21], ascribed mainly to their rumen biohydrogenation and to their poor intestinal digestibility [27,28]. The effectiveness of dietary modulation of the milk FA profile can be increased through technological treatments aimed at protecting dietary lipids from ruminal biohydrogenation, such as emulsification [29], encapsulation [30], or supplementation with calcium salts [31]. Some authors hypothesize that goats and ewes, due to their higher rumen transit compared to cows, would be subjected to a lower biohydrogenation rate [32,33]. Data comparing FA transfer efficiency of goats and cows are inconsistent among different studies [6,34,35].

The effects of dietary lipid supplementation on animal production [21,36,37] and reproductive performance [38,39,40] are widely documented. On the other hand, there are fewer references to the effects of dietary lipids on the health status of dairy ruminants. This review will attempt to summarize current knowledge about lipid supplementation in dairy goats, with reference to the effect on lipidic metabolism and immune response.

## 2. Fat Supplementation and Lipid Metabolism

In the scientific literature it is possible to find a conspicuous number of references about the effects of dietary lipids on ruminant metabolism. Most of the studies, however, are more focused on lipid metabolism of the mammary gland, in order to identify the consequences in terms of production. Long-chain PUFAs (EPA and DHA) at the ruminal level are able to disrupt the biohydrogenation process, inhibiting ruminal conversion to stearic acid [41,42]. The effect of EPA and DHA on rumen lipid metabolism is of considerable interest, as it can be exploited to modify the acidic profile of the milk. In an experimental trial conducted by Póti et al. [22], the administration of marine algae for a period of 16 days markedly modified the acidic profile of goat milk, with an increased concentration of PUFAs, MUFAs, and n-3 FAs, while reducing levels of SFAs. The shift in the rumen microbial population induced by PUFA supplementation, and the consequent increase in trans-intermediates of the linolenic acid biohydrogenation, also affects milk conjugated linoleic acid (CLA) content [43,44,45]. Beneficial effects on human health are attributed to CLAs, such as protection from cancer and obesity, as well as anti-atherogenic and anti-diabetic properties [46]. Despite all these positive effects on the acidic composition of milk, the activity of PUFAs at the ruminal level is also considered to be the main cause of the reduction in the fat content of milk, a phenomenon often detected following fat supplementation [20,21]. Several trans-CLA reaction intermediates resulting from the incomplete ruminal biohydrogenation of FA are supposed to inhibit the mammary synthesis of FAs. These trans-CLA intermediates exhibit anti-lipogenic effects of variable magnitudes. In particular, trans10-cis12-CLA has been demonstrated to be partly responsible for milk fat depression (MFD) in dairy cows [47,48]. In a study by Zhang et al. [49], trans10-cis12-CLA downregulated the expression of various genes involved in de novo FA synthesis (*FASN, SCD1, ACACA*) in caprine mammary gland epithelium in vitro. The reason behind such an effect are still to be clarified. In a recent study by Vargas-Bello-Perez et al. [50] on goat mammary cells, saturated long-chain fatty acids, in particular stearic acid, were able to modulate milk fat synthesis partly via peroxisome proliferator-activated receptor (PPAR)*γ*, while PPAR*γ* had an indirect role on PUFA regulation of milk fat-related genes: in particular it had a permitting effect for trans10-cis12-CLA and a blocking effect for DHA. Of note, other studies seem to indicate that trans10-cis12-CLA plays a less important role in dairy goats supplemented with fish oil than in dairy cows [35]. Whether this finding depends on differences in the responsiveness of the two species or species-specific lipogenic genes pathways is still unclear. Bernard et al. [51] found that goats were susceptible to MFD when supplemented fish oil but not sunflower oil, unlike cows in which milk fat content dropped in response to both treatments. Since various reports pointed out the relatively low sensitivity of the caprine species to the anti-lipogenic effects of ruminal biohydrogenation intermediates [35,52], fish oil-induced MFD experienced by goats in the latter study could be attributed to mechanisms other than direct trans-intermediate activity. An hypothesis in this regard takes into account the Δ9-desaturase-mediated conversion of milk C18:0 (positively related to milk fat content in goats) to cis-9 C18:1, in order to reduce the milk fat fluidity to adequate levels for milk secretion [53,54,55]. This theory seems to find confirmation in the study conducted by Toral et al. [35], who recorded a significant decrease in milk C18:0 and cis9 C18:1 content and an increase in cis-9 C18:1/C18:0 ratio in both cows and goats when supplemented with fish oil. In spite of this, dietary C18:0 supplementation proved ineffective in mitigating fish oil-induced MFD in dairy ewes [56]. Moreover, Bernard et al. [51] underlined that lipogenic genetic pathway in mammary gland differed between the bovine and caprine species. In line with the latter finding are those of Suárez-Vega et al. [57], who evidenced how MFD induced by CLA supplementation was characterized by a wider downregulation of adipogenic genes in dairy ewe mammary glands than fish oil-induced ones; moreover, CLA-induced MFD but not fish oil-induced MFD affected genes involved not only in FA synthesis, but also in immune response, as evidenced by the milk somatic cell transcriptome analysis. In this regard, Zhang et al. [49] found that trans10-cis12-CLA increased the concentration of lipid droplets in caprine mammary gland cells. As discussed later on, an increase in lipid droplet content in the goat leukocyte cytoplasm is associated with the activation of immune and inflammatory mechanisms [58].

A more limited number of studies are aimed at evaluating the effects of fat supplementation on lipid metabolism in other organs and tissues. Fat supplementation seems indeed to influence lipid metabolism not only in the mammary gland but also in other organs of primary importance for animal health, such as liver and adipose tissue.

In an experimental study conducted on 18 dairy goats, dietary supplementation with saturated FA (hydrogenated palm oil) during the transition period proved effective in altering hepatic gene expression [43]. In particular, palm oil treatment caused an increase in peroxisome proliferator-activated receptor α (*PPARα*) mRNA levels compared to a fish oil and basal diet. PPARα is a ligand-regulated nuclear receptor that, upon activation by binding with specific ligands, increases the transcription of genes involved in various lipid metabolism processes such as peroxisomal FA uptake and oxidation, FA trafficking, and lipolysis [59]. Since it has been proved that PPARα can modulate its own gene expression [60], it could be inferred that SFA supplementation can trigger PPAR biological activity. In vitro studies on bovine kidney cells were also effective in proving stearate and palmitate are strong activators of PPARα target genes [61]. The greater responsiveness of ruminant PPARα to SFAs is in contrast to data obtained from non-ruminant studies [62] and seems to be related to differences in spatial structure and electrostatic charge of their ligand binding domains [63]; this could reflect a species-specific teleological adaptative mechanism in response to the composition of non-esterified fatty acids (NEFAs: the major natural ligands and activators of PPARα), which in ruminant species are mainly composed of SFAs [63]. This speculation seems to be in agreement with the results obtained by Agazzi et al. [43], who recorded an increase in the mRNA levels of various PPARα target genes involved in FA synthesis (*ACACA*) and intramithochondrial transportation (*ACADVL, CPT 1A*) in dairy goats supplemented with palm oil. Nonetheless, other studies pointed out how UFA supplementation—compared to untreated basal diet—is still able to induce heighten *PPARα* gene expression in goat liver [64]. In this regard, attention should also be paid to the dietary n-6 to n-3 ratio, since higher proportions of n-6 were found to downregulate *PPARα* and *PPARγ* gene expression in goat muscle tissue [65]. In any case, mixed results were obtained by evaluating the expression response of PPARα in to n-3 PUFA supplement in other goat tissues, being able to upregulate it in muscle tissue [58] but proving to be ineffective in adipose tissue [66,67].

In a recent study conducted by Invernizzi et al. [68], liver and adipose tissue histological samples from goats fed different FA sources were collected and evaluated. The trial comprised 23 goats in their transition period—from −21 to 21 days in milk (DIM)—divided into three groups fed a basal diet, a diet supplemented with strongly saturated FA (stearate), or a diet supplemented with unsaturated FA (fish oil). Adipose tissue histology revealed that the reduction of adipocyte diameter (index of lipomobilization) was confined to a more limited time frame in the fish oil-fed group (from −7 to 7 DIM) compared to the other two groups (from −7 to 21 DIM) (Table 1). On the other hand, hepatocytes of goats fed fish oil on day 21 had higher levels of vacuolated cells (a marker of mild to severe cell suffering derived from lipomobilization) compared to the control and stearate group. Absence of necrosis foci and pyknotic nuclei implies a higher degree of chronic adaptation to lipomobilization compared to bovine species. The adaptation mechanism seemed to reach its highest in the fish oil group, as supported by the more gradual increase in suffering hepatocytes. In addition to histology, biochemical analysis was also performed and evidenced higher glycemic levels in both stearate- and fish oil-fed groups than on control group, validating the effectiveness of the widespread practice of supplementing fats during the peripartum to support the energy balance [69,70].

In subsequent studies, the same tissue samples were subjected to molecular analysis to investigate the expression of genes involved in lipid metabolism [71,72]. In hepatic parenchyma, stearate-fed animals exhibited a marked upregulation of the stearoyl-CoA desaturase (*SCD*) gene in the post-partum, a trend opposite to that showed by the other two groups. *SCD* encodes for stearoyl-CoA desaturase, a Δ9 desaturase analogous to that found in the mammary gland, thanks to which a small part of FAs exposed to ruminal biohydrogenation are dehydrogenated. Moreover, two genes encoding for enzymes involved in peroxisomal FA β-oxidation—*ACOX1* and *ACAA1*—were upregulated pre-partum in both stearate- and fish oil-fed groups, with the effect lasting longer in the fish oil group [72]. In adipose tissue, the mRNA levels of the *LPIN1* gene were heightened in both the experimental groups during the post-partum, in accordance with the findings of other studies [73]. This effect was more pronounced in the group supplemented with PUFAs. *LPIN1* is a gene encoding for lipin-1, a protein found in various anatomic district and with different functions closely related to lipid metabolism. Lipin-1 can modulate lipomobilization through its activity: in response to high intracellular lipid content, lipin-1 acts as a phosphatidate phosphatase (PAP) enzyme, converting phosphatidic acid into diacylglycerol and providing substrate for the synthesis of all classes of lipids while reducing free FA levels [74]. In addition to its PAP activity, lipin-1 also act as a co-binder of PPARα [75], promoting hepatic FA β-oxidation, and exhibits pro-inflammatory activity [76]. By this view, the overexpression of *LPIN1* in adipose tissue of goats fed fish oil during the two weeks around parturition is consistent with what was found during hepatic histological analysis by Invernizzi et al. [68], who reported a more restrained level of lipid infiltration in these animals. Finally, *FASN* expression was downregulated in both dietary treatments from –7 days before kidding to 7 days in milk. *FASN* is the gene encoding for fatty acid synthase, and its reduction could be suggestive of decreased lipogenesis.

There is convincing evidence in caprine species also that maternal dietary fat supplementation during gestation and lactation can influence the metabolism of the offspring in different ways, such as altering their ruminal microbiome and adipose tissue proteome [77,78]. Some findings seem also to imply that the modifications induced by FAs on lipid metabolism could have a certain degree of heritability thanks to epigenetic mechanisms [79,80].

## 3. Fat Supplementation and Immune Response

The ability of specific FAs to modulate immune function and associated inflammatory processes has long been known and reported in scientific literature [81,82]. In general, it is currently accepted that dietary SFAs are related to increased risk of cardiovascular and coronary heart disease (CVD/CHD). The etiopathogenetic mechanism behind the association between SFAs and CVD/CHD is likely to be found in their tendency to raise blood low-density lipoprotein cholesterol levels [83]. Despite this, dairy SFAs seem to have a smaller to neutral association with CVD/CHD risk compared to other dietary sources [84,85], and there is evidence that at least some of them—including several CLAs—do not lead to increased serum cholesterol values [86,87].

On the other hand, PUFA consumption—and in particular n-3 PUFAs—is considered beneficial to human health [88]. Observational studies support this thesis, underlining the beneficial effects associated with a lower SFA and a proportionally higher PUFA dietary regime [89,90].

n-3 PUFAs exert an anti-inflammatory action through different molecular mechanisms. Until a few years ago it was widely believed that, thanks to their double bonds, PUFAs were able to fluidify and stabilize cell membranes, avoiding activation of the arachidonic acid cascade and improving cellular functionality [91]. To date, this hypothesis is considered controversial due to contradictory results [92]. On the other hand, n-3 PUFA integration into the phospholipidic double layer is able to modify the composition of lipid raft, affecting the activity of the protein receptors associated [93]. Among these proteins potentially modulated by n-3 PUFAs, some are involved in the activation of both innate and acquired immune response, such as Major Histocompatibility Complex (MHC) [94] and Toll-Like-Receptor 4 [95]. The effect of n-3 PUFAs on TLR-4, in particular, would also affect inflammation on a secondary level, given the role played by TLR-4 in the activation of inflammatory gene pathways; in this way, n-3 PUFAs are able to indirectly silence genes involved in the activation of the inflammatory cascade [96]. In addition, n-3 PUFAs also directly mitigate pro-inflammatory signaling of some transcription factors, including nuclear factor kappa-light-chain-enhancer of activated B cells (NFκB) [97,98]. In recent years, growing interest has surrounded a series of bioactive oxygenated metabolites of PUFAs, namely oxylipins. The most well-known class of oxylipins is of pro-inflammatory eicosanoids derived from arachidonic acid. Oxylipins act as signaling molecules, with a pro-or anti-inflammatory action based on the PUFA line from which they originate. Simplifying, oxylipins derived from n-6 PUFAs (arachidonic acid, linoleic acid) exhibit pro-inflammatory capacity, while n-3-derived oxyplins are mainly anti-inflammatory [99]. Anti-inflammatory n-3-derived oxylipins (resolvins, protectins, marensins) are called specialized pro-resolving mediators (SPM) by means of their ability to resolve acute inflammatory states, protecting the body from various inflammatory and metabolic insults and contributing to tissue regeneration [100]. Despite the PUFA precursor line being a factor of primary importance in defining the pro-or anti-inflammatory oxylipin activity, recent acquisitions on the topic have clarified how the biological activity of oxylipins depends on a more than previously thought complex biosynthesis network modulated by multiple factors, comprising the amount of PUFA precursors available and the tissue-specific enzymatic activity [101]. Moreover, emerging evidence shows how some of the anti-inflammatory effects of PUFAs could be mediated by epigenetic mechanism [102], opening up fascinating scenarios about their putative inter-generational effects.

The anti-inflammatory and immunomodulatory activity of n-3 is exploited in the field of animal science to obtain products of animal origin characterized by specific nutraceutical properties, able to provide a substantial contribution to protecting human health from pathological states. Nevertheless, supplementing n-3 in the diet has also proved to be able to directly and positively influence the health status of animals in their production phase.

Studies have been conducted in the past to assess the immunomodulatory effect of dietary n-3 PUFAs in goat, both in vivo and in vitro.

Goats fed a fish oil-supplemented diet during the transition period manifested a greater than control group skin reaction to a phytohaemagglutinin (PHA)-challenge skin test after kidding [103,104], demonstrating a higher level of cell-mediated immunity. Consistently, higher levels of circulating lymphocytes were found in supplemented animals, indicating that fish oil was effective in contrasting the drop of lymphocytes concentration and functionality occurring in the peripartum [105,106]. Supplementing n-3 PUFA to dairy goats altered the CD4+/CD8+ lymphocyte ratio during peripartum, favoring the CD4+ subset (T-helper lymphocytes) [107]. By contrast, the administration of corn oil rich in n-6 PUFAs reduced the surface expression of CD49d (α-4 integrin)—a subunit of lymphocytes homing receptors that modulate tissue-specific cell trafficking—in lymphocytes of male castrated goats [108].

In vitro studies evaluated leukocyte functionality in fat-supplemented goats. Pisani et al. [109] found that neutrophils isolated from dairy goats produced less reactive oxygen species (ROS) (oxidative burst) when exposed to various doses (25, 100, 200 μM) of DHA, but not of EPA. Results also pointed out how even low doses (25 μM) were strongly effective in stimulating neutrophil phagocytic activity. This data implies a positive effect of n-3 PUFAs on neutrophilic activity, favoring phagocytosis while reducing extracellular release of ROS, that in situations of chronic and dysregulated inflammation can be potentially harmful for the host tissues. Similar results were also found by Lecchi et al. [110] in goat monocyte-isolated cells (Figure 1). A dose/effect dependence was identified, since the highest levels of phagocytic activity were stimulated by low-to-mid doses of EPA (25–100 μM; >47%) and only mid doses of DHA (100 μM; 25%). Higher doses of both EPA and DHA are not only shown to be ineffective on cultured monocyte cells, but also detrimental, as evidenced by the increase in caspase-7 and caspase-3 pro-apoptotic activity [58]. Through the same study, treatment of caprine monocytes with EPA and DHA caused a dose-dependent increase in intracytoplasmic lipid droplets. High-dose treatment (>200 μM) with EPA also significantly upregulated mRNA of *PLIN3*, a gene encoding for a protein of the perilipin family (perilipin-3) which acts as a lipid droplet-associated protein contributing to lipid droplets formation [111]. Recent developments in research have expanded knowledge about lipid droplets, which are now recognized as structures actively involved in inflammatory and immune processes [112].

In a study of Farina et al. [113] on adipose tissue, samples collected by goats subjected to different types of fat supplementation (stearate vs. fish oil vs. control) underwent molecular analysis aimed at identifying variations in the expression of genes involved in adipose inflammation. The study did not evidence any effect of FA supplementation on the inflammatory gene pathway, except for a significant decrease of interleukin 6 (*IL6*) expression in the stearate group compared to the control. Inflammation-related gene expression was instead affected by time around parturition, with a firm overexpression after kidding followed by a gradual decline at 21 days after parturition. This effect can be interpreted as a further confirmation of the differences in the bovine and caprine metabolic processes, the latter of which would seem to be characterized by a shorter duration and lower intensity of the modifications of the metabolic and immune functionality typical of the peripartum. In the study, quantitative PCR was also applied to evaluate the expression of selected target microRNA(miRNA). As for mRNA, none of the selected miRNAs showed diet-dependent variations in expression levels, but one miRNA associated with immune cell infiltration (miR-155) was overexpressed in the two weeks around kidding. Despite none of miRNAs directly associated with inflammation being overexpressed, adipose tissue immune cell infiltration is strongly associated with lipolysis and adipose inflammation phenomena commonly displayed throughout the transition period [114]. Since to our knowledge this is the first report of miRNA analysis performed on transition dairy goats’ adipose tissue, whether these findings are attributable to a real lack of effect or to the choice of target miRNA still needs to be clarified.

## 4. Conclusions

Fat supplementation of the dairy goat diet is not considered a mere strategy to increase the energetic level of the rations. Continuous advances in the knowledge of ruminant lipid metabolism, coupled with the awareness of the multiple actions that specific FAs have on it, have led to their application in order to modulate the specific characteristics of animal production, such as increasing healthy UFA and CLA content. When opting to integrate fats into the diet in order to enrich the acidic profile of the milk, particular attention should be paid to the choice of lipid sources. However, PUFA supplementation has also shown its inhibitory effects on ruminal metabolism, with the consequent risk of a drop in milk fat content (MFD). Even if in goats this effect is less pronounced than in cows and sheep, to increase PUFA’s transfer efficiency in milk, a good strategy could be to use rumen-protected or rumen inert fat supplements. To date, it is well known how adipose tissue does not only represent a simple reserve of body fat, but it is also an endocrinally and paracrinally active tissue capable of producing and secreting numerous lipid mediators. Similarly, dietary FAs exhibit modulatory capacities with respect to metabolic, inflammatory, and immune processes, through direct action on receptors and transcription factors as well as through their bioactive oxygenated metabolites. Dietary FAs exert a modulating action on the expression of several genes involved in lipid metabolism. These effects are ubiquitous, occurring in various body tissues. As already mentioned, at the mammary gland level the major implications of dietary FAs action are of a productive nature, with the potential to modify the concentration and the acidic profile of milk fat. At the hepatic level the supplementation of fat with the diet is correlated to variations in the expression of a pool of genes involved in lipid metabolism. These effects seem to be more pronounced in the postpartum period, probably due to the challenge represented by the metabolic changes that affect the first days in milk. Attention should be paid also in this case to the class of FAs supplied, since SFAs and n-3 and n-6 PUFAs showed different affinity and effects on target genes. In general, SFAs and n-3 PUFAs are both PPARα ligands, with SFAs proven to be more effective ligands while n-6 PUFAs act as PPARα inhibitor. In addition, n-3 PUFA supplementation was associated with higher expression of various genes involved in β-oxidation and lipid mobilization in both hepatic and adipose tissue. Results from histological analysis of liver tissue consistently showed lower and more gradual fatty infiltration in fat-supplemented goats, particularly in those which received n-3 PUFAs. Regarding the immune modulatory activity of dietary fats, signs of improved immune functionality in n-3 PUFA-supplemented goats were clear. It can be assumed a beneficial effect of long-chain PUFAs on the host defense from non-specific immune stimuli, an aspect that—coupled with the energy supply provided by lipids—can be particularly useful during metabolic and immune stress conditions such as those that characterize the peripartum period. Further studies should be carried out to investigate the relationship between dietary fats and lipid droplets, intracellular organelles that are provoking a growing interest in relation to their newfound active role in immune and inflammatory processes. Molecular analysis studies seem to be more elusive, proving to be ineffective in identifying any correlations between dietary supplement and inflammation- and immunity-related genes. Nonetheless, comparison with the most numerous analogous studies on dairy cow allowed the identification of discrepancies between the two species in the time trend and intensity of inflammation, lipomobilization, and immune modifications that accompany parturition. In general, investigations on genetic pathway modifications can still prove extremely useful to increase our knowledge on the complex ensemble of homeorhetic mechanism put into action by ruminants in response to the metabolic and immune challenge they go through during the peripartum. In this regard, particular mention should be given to the studies aimed at evaluating goat miRNAs. Given the recent advances in understanding the epigenetic mechanisms underlying the gene silencing functions of miRNAs, these studies can pave the way for new insights on the nutritional modulation of transcriptomics (i.e., nutrigenomics).

## Figures and Tables

**Figure 1 animals-09-00917-f001:**
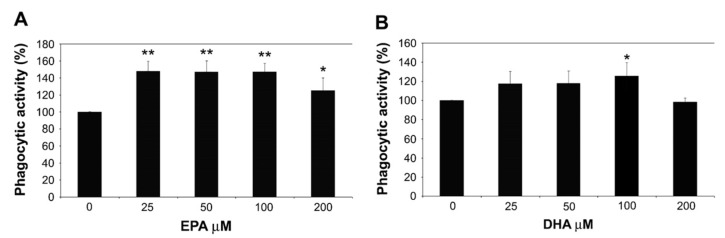
Effects of eicosapentaenoic acid (EPA (**A**) and docosahexaenoic acid (DHA) (**B**) on the phagocytic activity of caprine monocytes (*p* < 0.05, ** *p* < 0.01; [110], modified).

**Table 1 animals-09-00917-t001:** Mean adipocyte area (μm^2^) of subcutaneous adipose tissue of dairy goats fed with a basal diet (C), a diet supplemented with fish oil (FO), or stearic acid (ST) [68].

Time	Treatment ^1^						*p*-Value ^2^		
	C	SD	FO	SD	ST	SD	Trt	Time	Trt × Time
−7	3200.00 ^d^	1002	2877.69 ^d^	876	3088.80 ^d^	1010	0.60	<0.01	<0.01
7	1970.44 ^e^	523	1801.80 ^e^	548	2156.33 ^e^	718			
21	1157.74 ^b,f^	216	1851.85 ^a,e^	638	1066.66 ^b,f^	269			

^a,b^ Values within each row with different superscripts are significantly different (*p* < 0.05). ^d–f^ Values within each column with different superscripts are significantly different (*p* < 0.05). ^1^ SD, standard deviation ^2^ Trt, treatment effect.

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
