# Peer review of "Effects of Fat Supplementation in Dairy Goats on Lipid Metabolism and Health Status"

_animals, 2019, doi:10.3390/ani9110917_

Round 1
Reviewer 1 Report
Reviewer 1
The theme chosen in this article is meaningful. It has a good guiding significance to link dietary fat supply with fat metabolism (especially unsaturated fatty acids) and goat health, and provided some guidance information for human diet. In addition, the idea is clear and the arguments are adequate in overall. However, there are still some points in this manuscript that need to be corrected or supplemented before publishing.
Line 57-58: “technological treatments” belong to “nutritional factors” or “feed processing”? Technological treatments affect the diet nutrient, but I do not agree with categorizing technological treatments as a nutritional factor.
Line 113-114: Could you supplement related information about why the MFD showed different response to fish oil and sunflower oil?
Line 131-132: Explain why the effects of the two oils are different or provide relevant speculation.
Line 132-134: Supplement Reference Information.
Line 134-144: Please not just list the relevant references information. It is necessary to compare the differences and similarities of these cited papers in order to provide more accurate reference information for readers.
Line 150-151: Please provide the full name of DIM.
Line 188-189: What’s the influence of maternal dietary fat supplementation on offspring metabolism?
Line 287-343: The conclusion part is too long to get the key point of this paper. Please simplify the conclusion section, and move some critical statements to “Fat supplementation and lipid metabolism” and “Fat supplementation and immune response” parts.
Reviewer 2 Report
Savoini et al. summarized current knowledges about lipid supplementation in dairy goats, with reference to the effect on lipidic metabolism and immune response. The manuscript has a good amount of citations to the references. The article is clearly written and the illustrations are very appropriate. This review article is worthy of publication.
Only a few typos were found:
Line 94, trans- intermediate; line 121, Zhang et al.,; line 269, [106],;
In references, some journal abbreviation name were wrong. For example, ref 1, 2, 4, 7,22, and 107.
